# Travel time to care does not affect survival for patients with colorectal cancer in northern Sweden: A data linkage study from the Risk North database

Olle Sjöström[1]*, Anna M. Dahlin[1], Gustav Silander[1], Ingvar Syk[2], Beatrice Melin[1], Barbro Numan Hellquist[1]

1 Department of Radiation Sciences, Oncology, Umeå University, Umeå, Sweden, 2 Department of Surgery, Skåne University Hospital, Lund University, Malmö, Sweden

☯ These authors contributed equally to this work.

* olle.sjostrom@regionjh.se

**Data Availability Statement:** The data are not publicly available due to restrictions e.g. their containing information that could compromise the

## Abstract

### Introduction

Numerous prior studies, even from countries with free access to care, have associated long travel time to care with poor survival in patients with colorectal cancer.

### Methods

This is a data-linkage study of all 3718 patients with colorectal cancer, diagnosed between 2007 and 2013 in Northern Sweden, one of the most sparsely populated areas in Europe. Travel time to nearest hospital was calculated based on GPS coordinates and multivariable Cox regression was used to analyse possible associations between travel time and cause-specific survival.

### Results

No association between travel time and survival was observed, either in univariable analysis (colon HR 1.00 [95% CI 0.998–1.003]; rectal HR 0.998; [95% CI 0.995–1.002]) or in multi-variable Cox regression analysis (colon HR 0.999 [95% CI 0.997–1.002]; rectal HR 0.997 [95% CI 0.992–1.002]).

### Conclusions

In contrast to most other studies, no association between travel time and colorectal cancer survival was found; despite that longer travel time was associated with known risk factors for poorer outcome. In the Swedish health care setting, travel time does not appear to represent a barrier to care or to negatively influence outcomes.

privacy of research participants. Although the identity of the research objects are protected by a code key, the GPS coordinates could be used to identify individual participants. Data can be made available from Department of Radiation Sciences, Oncology, Umeå University, Umeå, Sweden - if the request is approved by the Ethics Committee. Contact via rccnorr@umu.se.

**Funding:** This study was funded with grants from Regional Cancer Centre North (RCC Norr)BNH and AD, the County Councils of Västerbotten (ALF Grants) GS, OS and Jämtland-Härjedalen OS. The funders had no role in study design, data collection and analysis, decision to publish, or preparation of the manuscript.

**Competing interests:** The authors have declared that no competing interests exist.

## Introduction

Long travel distance and long travel time to care have been linked to poor outcomes in patients with malignant diseases [1–3]. These findings might reflect barriers to access of care, possibly increasing risk not only for delayed diagnosis and suboptimal treatment, but also for lower patient participation in screening programs and follow-up care.

Several studies have found evidence of poor survival in patients with colorectal cancer who reside in rural areas [3–5]. In many of these studies, a more advanced stage at diagnosis was found in patients who travelled long distances to receive care, findings that suggest stage at diagnosis is a mediator to an unfavourable outcome [4, 5]. In addition, advanced disease stage at diagnosis increases the risk for emergency surgery, which is an independent risk factor for poor survival in colorectal cancer [6, 7]. There is also evidence that provisions for surgical and oncological treatments differ between rural and urban areas [4, 8]. However, differences in colorectal cancer survival attributed to distance to care might in part be explained by sociodemographic differences between rural and urban populations.

In many areas in Europe and the U.S., the rural population is older and has a lower socioeconomic status compared with urban populations [9, 10]. Both older age and low socioeconomic status have been associated with poorer prognosis in colorectal cancer [11, 12]. In addition, there may be differences between rural and urban areas in the proportion of persons living alone, a factor that has been shown to adversely affect outcomes in colorectal cancer [11–13].

Recently, we published a study that found lower survival for patients with colon cancer in the Northern Healthcare Region in Sweden, compared with the rest of the country [12]. The Northern Healthcare Region has approximately 900 000 inhabitants living in an area of 224 000 $km^2$, resulting in a population density of only 4 pop./$km^2$. (U.K. $\approx$ 260 pop./$km^2$, France $\approx$ 100 pop./$km^2$) [14, 15]. The Region includes many rural areas where patients have to travel long distances to reach the nearest hospital (Fig 1). In our earlier study, we proposed that on average longer distance to health care might contribute to the poor survival observed in the Northern Region. This study investigates whether travel time to the nearest hospital in the Northern Healthcare Region is associated with survival in patients diagnosed with colorectal cancer.

## Methods

This cohort study is based on information retrieved from the Risk North database, which includes all men and women diagnosed with colorectal cancer in the Northern Healthcare Region between January 1st 2007 and December 31st 2013. The Risk North database was constructed to study associations between sociodemographic factors, cancer management, and cancer survival in the Northern Healthcare Region of Sweden. To enable studies on cancer disparities between the Northern Health Care Region and the rest of the country, the database also includes data on patients residing in other Swedish regions. The Risk North project and database has been described in detail earlier [12].

Briefly, the database was generated by means of individual level record linkages between three national cancer quality registries (the Colorectal Cancer Registry, the Oesophageal and Gastric Cancer Registry, and the Brain Tumour Registry) and other demographic and health care registries (S1 Fig). Combining data from the same individual from different data sources is possible because the Swedish government issues a personal identity number to all individuals in Sweden at birth or time of permanent residency. This identity number is used to track the use of all health care services.

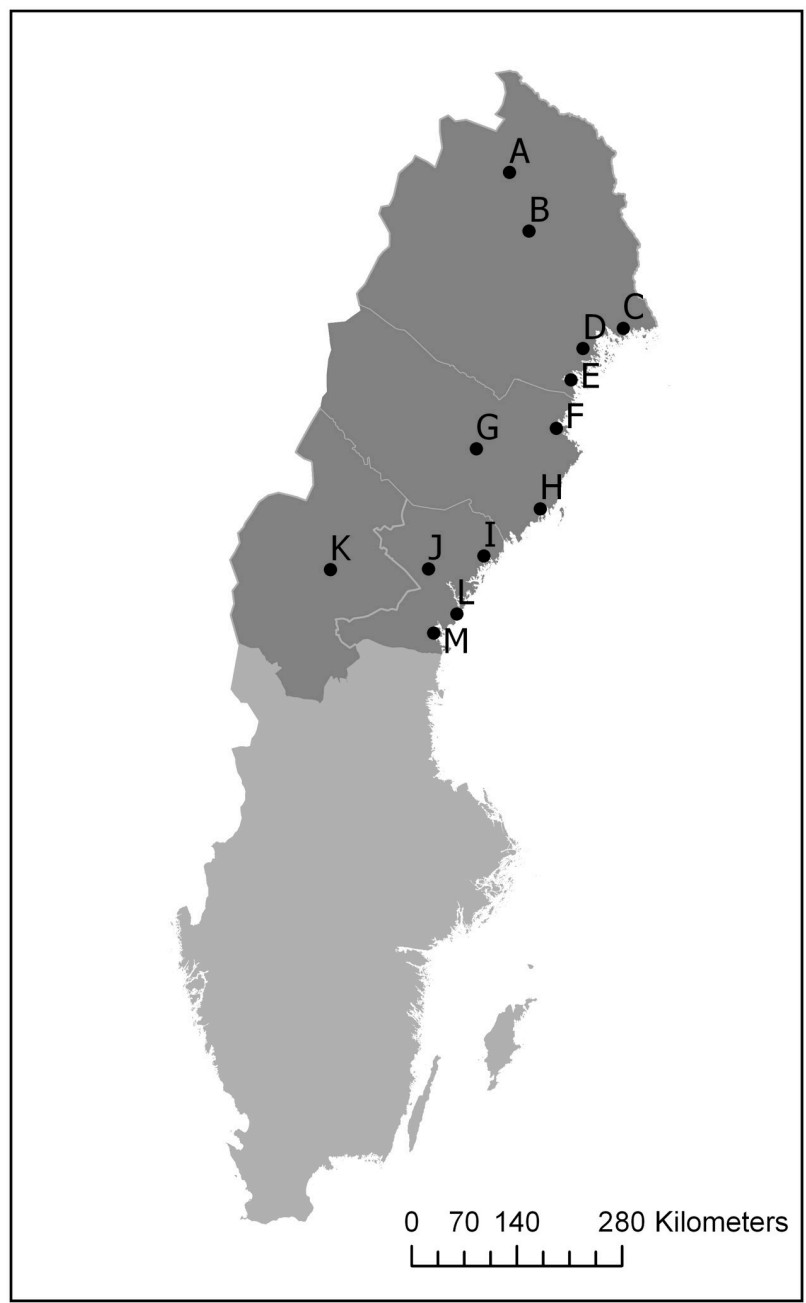

**Fig 1. Map of the Northern Health Care Region of Sweden with hospitals.** Hospitals: A Kiruna, B Gällivare, C Kalix, D Sunderbyn, E Piteå, F Skellefteå, G Lycksele, H Umeå (University Hospital), I Örnsköldsvik, J Sollefteå, L Härnösand, M Sundsvall, K Östersund. (Source: The Swedish Mapping Authority (Lantmäteriet), available according to open data license Creative Commons, CC0).

## Original sources of data in Risk North used in the present study

The data in the Risk North database used in the present study originates from five different registries: Swedish Colorectal Cancer Registry (SCRCR), Cause of Death Register (CDR), Geography Database (GD), Total Population Register (TPR), and Longitudinal Integration Database for Health Insurance and Labour Market Studies (LISA).

**Swedish Colorectal Cancer Registry (SCRCR).** The SCRCR is a national quality registry with the primary purpose to monitor management and outcomes. To ensure that no patients are missed, the SCRCR is cross-matched with the National Cancer Registry, resulting in 98% level of completeness [16].

Since the start of the registry in 1995, data from the SCRCR have been used in many scientific studies [17, 18].

The present study uses the following variables from the SCRCR: hospital, age, gender, tumour stage, elective surgery, and emergency surgery. In the SCRCR, tumour stage is recorded according to the Union for International Cancer Control (UICC) TNM classification 7th edition. Stage is primarily based on histopathological staging of surgical specimens (pTNM), if histopathology is missing—clinical staging (imaging) is used (cTNM). Emergency surgery is defined as an operation during an unplanned admission as the result of an acute medical condition such as bowel obstruction or perforation.

**The Cause of Death Registry (CDR).** The CDR, part of the National Board of Health and Welfare, includes date of and causes of death for every diseased individual in Sweden. The level of completeness is very high and in patients with malignant disease the validity of the recorded cause of death has been estimated to be around 90% [19]. In this study, we used the CDR to obtain information on date and cause of death. Cause-specific death was defined as a death where colon or rectal cancer was listed as the main cause or one of first two contributing causes.

**Geography Database (GD).** The GD links personal identity number to GPS coordinates of the patient's registered address. The coordinates of the patient's address the year before the diagnosis was used to calculate travel time to hospital. The precision of the coordinates is 250 x 250 m in urban areas and 1000 x 1000 m in rural areas.

**The Total Population Registry (TPR).** The TPR contains information on age, sex, and registered addresses for the entire Swedish population. We used the TPR to determine whether a patient was living alone or co-habiting the year before diagnosis.

**Longitudinal Integration Database for Health Insurance and Labour Market Studies (LISA).** LISA merges several registries covering health insurance and the Swedish labour market and contains individual information on socioeconomic factors such as income and educational level. Previous studies have suggested that education represents the best predictor for socioeconomic-related health outcomes [20]. Consequently, we used the highest level of education as a measure of socioeconomic status. Level of education was defined as low (up to nine years of compulsory school), middle (2 to 3 years of secondary education), or high (university).

## Statistical analysis

In the analyses of possible associations between travel time and colorectal cancer survival, several potential confounders or mediators were considered: age, socioeconomic status, co-habiting status, tumour stage, proportion of emergency operations, and proportion of surgical resection. To test for proposed associations, we used two-sided between-subject Student's t-test for continuous parametric variables and Spearman's test for non-parametric ranked variables ($\alpha = 0.05$).

**Assessments of travel time.** We measured travel time by car from the patient's home address to the nearest hospital with facilities to diagnose and stage colorectal cancer (i.e., endoscopy and/or radiology department). For operated patients, we also measured travel time to the operating hospital registered in the SCRCR. For the location of the hospitals, see Fig 1. Coordinates for the hospitals were retrieved from a Swedish search engine, Eniro.se [21].

All travel time calculations were performed with ArcGIS® Pro (2.1.2) and ArcGIS online (Esri, Redlands, CA, USA). For each hospital in the Northern Region, we created 10-minute interval drive time areas: 0–10; >10–20; >20–30; . . .; >290–300. We used default settings of the 'Rural driving time mode' in ArcGIS®, as traffic rarely hampers transportation in Northern Sweden. Individual driving times for each patient were then identified using the ArcGIS® tool 'Spatial Join'.

**Survival analysis.** Colorectal cancer-specific survival was defined as the time from date of diagnosis to date of death attributed to colorectal cancer. Patients were censored at time of death due to other causes, emigration abroad, or at the end of follow-up (31 December 2014).

In our main survival analysis, travel time was handled as a continuous variable, using the lowest value in the above-described 10-minute intervals.

In univariable analysis, Kaplan-Meier estimates were used to plot survival in patients with different travel times, and Cox regression analysis was used to estimate hazard ratios (HRs) of death with 95% CI for patients with different travel times.

For multivariable analysis, multiple Cox regression analysis was used to estimate hazard ratios (HRs) of death with 95% CI for patients with different travel times. This analysis was stratified by age (10-year groups) and gender and adjusted for stage, educational level, co-habiting status, and emergency surgery.

Patients were excluded if data were missing for any of the co-variables in the multivariable analysis, and consequently we excluded all non-operated patients (missing data on the variable operation).

In all Cox models, the proportional hazard assumption was tested.

We performed a sensitivity analysis of the results in the main multiple Cox regression by varying end-points, input variables, and adjusted factors.

We also performed an additional multiple Cox regression survival analysis, where we handled travel time as a categorical variable—comparing survival in patients traveling < 1 h vs. patients traveling >1 h.

We used R version 3.6.0 for the statistical analysis (R Core Team) [22].

## Ethics

The Regional Board of Ethics in Umeå approved the design of the Risk North database and the present research project—approval number: 2014/278-31. All colorectal cancer patients in Sweden are informed about registration in the SCRCR (i.e. the primary source of data in the present study) and an opt-out procedure for registration is used in the SCRCR. The data in the Risk North Database is not publicly available according to the Swedish data protection law. The patients did not provide informed written consent but all data were fully anonymised before access.

## Results

During the study period (2007–2013), 3721 men and women were diagnosed with colorectal cancer in the Northern Healthcare Region and registered in the SCRCR. Three patients were excluded because of missing geographical coordinates, thus the final study population included 3718 patients (Table 1). The mean travel time to nearest hospital was 23.85 minutes with standard deviation 33.67. About one-third (36.8%) of the patients had less than 10 minutes travel time to a hospital that provides care (Fig 2). The five-year cause-specific survival rate for all included colorectal patients was 64% (colon: 64%; rectal: 63%).

**Table 1. Distribution of sociodemographic and tumour characteristics in 3718 patients residing in the Northern Region and diagnosed with colorectal cancer between 2007 and 2013.**

|  | N (%) | Missing (%) | Total |
|---|---|---|---|
| **Gender** |  |  |  |
| Male | 1970(53.0) | 11(0.3) | 3718 |
| Female | 1737(47.0) |  |  |
| **Tumour location** |  |  |  |
| Colon | 2463(66.2) | 2(0.05) | 3718 |
| Rectum | 1253(33.8) |  |  |
| **Tumour stage** |  |  |  |
| I | 563(15.1) |  | 3718 |
| II | 1005(27.0) | 300(8.1) |  |
| III | 1063(28.6) |  |  |
| IV | 787(21.2) |  |  |
| **Surgery[1]** |  |  |  |
| Yes | 3248(87.4) | 0 | 3718 |
| No or missing data | 470(14.5) |  |  |
| **Educational level** |  |  |  |
| Low | 1545(41.6) | 18(0.005) | 3718 |
| Middle | 1509(40.6) |  |  |
| High | 646(17.4) |  |  |
| **Co-Habiting status** |  |  |  |
| Living Alone | 1577(42.4) | 0 | 3718 |
| Co-habiting | 2141(57.6) |  |  |

[1]Operation; Yes includes any operation recorded—curative or palliative, whereas No represents no surgery recorded or performed.

Missing data was < 1% for variables gender, tumour location, educational level and co-habiting status (Table 1). For tumour stage, 300 patients (8.1%) had missing data. Three patients were lost to follow-up in the survival analysis.

No of patients in each time interval:

$0 - 10$ 1371(37%) $> 10 - 20$ 827(22%) $> 20 - 30$ 338(9%) $> 30 - 40$ 284(8%)

$> 40 - 50$ 199(5%) $> 50 - 60$ 183(5%) $> 60 - 70$ 92(2%) $> 70 - 80$ 129(3%)

$> 80 - 90$ 44(1%) $> 90 - 100$ 29(1%) $> 100 - 110$ 22(1%) $> 110$ 110(3%)

## Patient characteristics and travel time

We found associations between longer travel time and older age (p = 0.044), lower educational level (socioeconomic status) (p < 0.001), and living alone (p = 0.003) (Table 2; S2–S4 Figs). No significant differences in tumour stage at diagnosis were found with respect to travel time to the nearest hospital (Table 2; S5 Fig) (p = 0.96). The risk for emergency surgery was not associated with travel time, or the proportion of non-operated patients (p = 0.767) (Table 2 and S6 Fig).

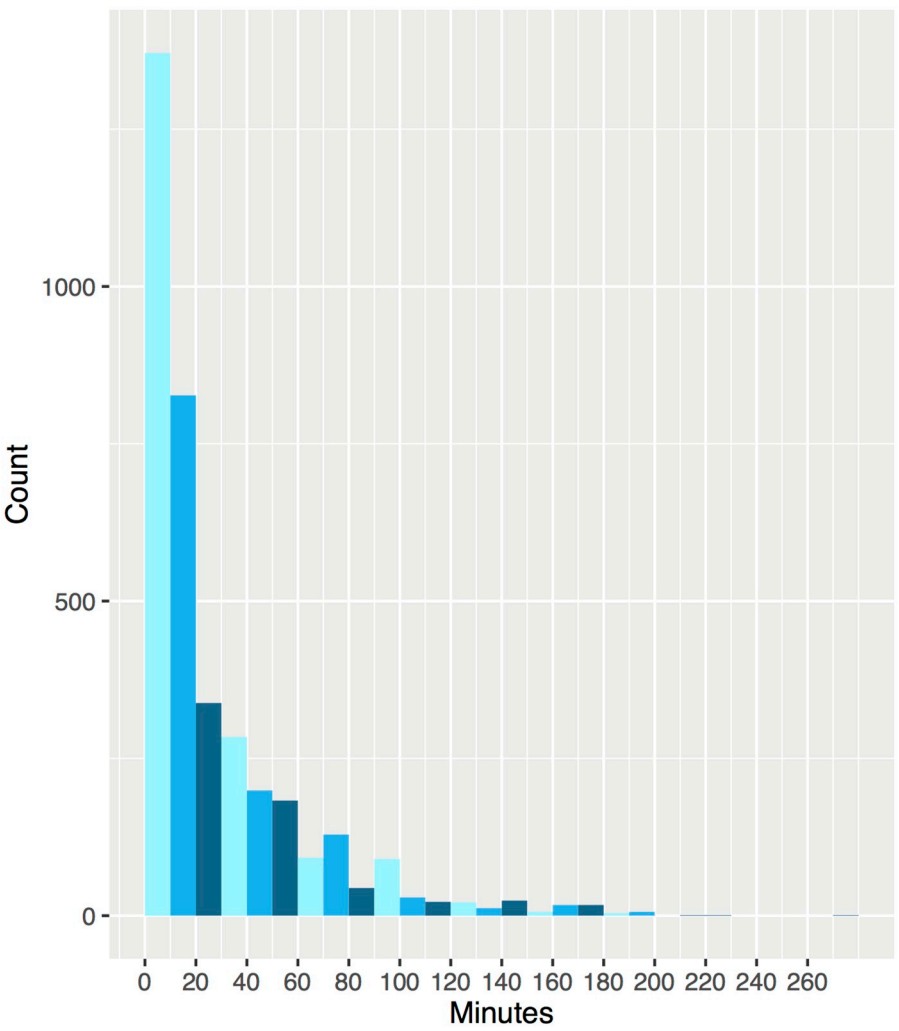

**Fig 2. Distribution of travel time (minutes) to nearest hospital for patients with colorectal cancer residing in the Northern Region of Sweden.**

### Analysis of the impact of travel time on colorectal cancer survival

In the univariable analysis including all patients, there were no significant differences in cause-specific survival in relation to travel time to the nearest hospital for colon (HR 1.001; 95% CI 0.9985–1.003) or rectal cancer (HR 0.9983; 95% CI 0.9952–1.002).

Fig 3 graphically represents univariable survival by different travel times in a Kaplan-Meier plot.

The main multiple Cox regression analysis (restricted to operated patients) revealed no statistically significant differences in cause-specific survival in relation to travel time. However, survival was significantly lower in colon cancer patients living alone and in patients who had emergency surgery or advanced stage cancer at diagnosis (Table 3).

The stability of the results in the main multivariable regression survival analysis was assessed with a sensitivity analysis. In these analyses, the association between travel time and survival was examined in five other settings: 1) colorectal cancer survival handled as one entity rather than colon and rectal cancer separately; 2) over-all cancer survival rather than

**Table 2. The patient´s characteristic—Mean age, level of education, co-habiting status, tumour stage at diagnosis and type of surgery—Stratified by travel time to the nearest hospital.**

| Characteristics | Travel time (min) | | | | | | | | P-value |
|---|---|---|---|---|---|---|---|---|---|
| | **0–10** | **>10–20** | **>20–30** | **>30–40** | **>40–50** | **>50–60** | **>60–70** | **>70** | |
| **No. of patients** | 1371 | 827 | 338 | 284 | 199 | 183 | 92 | 424 | |
| **Mean age yrs** | 70.7 | 70.2 | 69.7 | 70.3 | 71.6 | 73.2 | 72.7 | 72.4 | 0.044[1] |
| **Level of education N (%)[2]** | | | | | | | | | |
| Low | 499(36) | 304(37) | 142(42) | 139(49) | 108(54) | 90(49) | 48(52) | 215(51) | <0.001[3] |
| Middle | 555(40) | 377(46) | 141(42) | 105(37) | 69(35) | 69(38) | 35(38) | 158(37) | |
| High | 312(23) | 144(17) | 51(15) | 39(14) | 20(10) | 24(13) | 8(9) | 48(11) | |
| Missing | | | | | 18 | | | | |
| **Co-habiting (%)[2]** | | | | | | | | | |
| Proportion of patients living alone | 609/1371 (44) | 315/827 (38) | 119/338 (35) | 83/284 (40) | 81/199 (42) | 38/183 (44) | 60/92 (41) | 272/424 (51) | 0.003[3] |
| **Tumour stage (%)[2]** | | | | | | | | | |
| I | 226(16) | 111(13) | 50(15) | 37(13) | 30(15) | 29(16) | 17(18) | 63(15) | 0.431[3] |
| II | 373(27) | 221(27) | 85(25) | 90(32) | 51(26) | 50(27) | 25(27) | 110(26) | |
| III | 383(28) | 244(30) | 101(30) | 68(24) | 62(31) | 47(26) | 26(28) | 132(31) | |
| IV | 282(21) | 181(22) | 73(22) | 66(23) | 41(21) | 48(26) | 14(15) | 82(19) | |
| Missing | 107(8) | 70(8) | 29(9) | 23(8) | 15(8) | 9(5) | 10(11) | 37(9) | |
| **Type of surgery (%)[2]** | | | | | | | | | |
| Elective | 962(70) | 575(70) | 241(71) | 195(69) | 131(66) | 127(69) | 65(71) | 309(73) | 0.767[3] |
| Emergency | 180(13) | 115(14) | 39(12) | 43(15) | 33(17) | 23(13) | 10(11) | 50(12) | |
| Not operated or missing | 229(17) | 137(17) | 58(17) | 46(16) | 35(18) | 33(18) | 17(18) | 65(15) | |

[1] Student's t-test.

[2] Percentages may not total 100 due to rounding.

[3] Spearman's test.

cause-specific survival; 3) cause-specific survival without adjusting for tumour stage and emergency operations; 4) travel time to the operating hospital rather than the nearest hospital; and 5) the exclusion of all patients with tumour stage IV.

In all five of these alternative models, the results remained robust: there was no association between travel time and survival (S1–S5 Tables).

In the additional multivariable regression survival analysis where travel time was handled as a categorical variable (survival for patients with travel time < 1 h was compared to patients traveling > 1 h) we found no association between travel time and survival, either for patients with colon cancer HR 0.92[95% CI 0.71–1.19] or rectal cancer HR 0.84 [95% CI 0.52–1.36] (S6 Table).

## Discussion

Longer travel time to the nearest hospital for patients with colorectal cancer was associated with older age, lower socioeconomic status, and living alone. However, we found no evidence of an association between travel time and stage at diagnosis or colorectal cancer survival.

Our study´s lack of an association between travel time to care and CRC survival is in contrast to the results of most other studies [2–5, 8, 23–25]. However, it is unclear to what extent previously observed associations between longer distance to care and poor colorectal cancer survival reflects access to care or confounding factors such as age, socioeconomic factors, and co-morbidity. One proposed mechanism for a direct distance–survival effect is late diagnosis due to less access to care, resulting in a more advanced tumour stage at diagnosis. In contrast

## Colon cancer

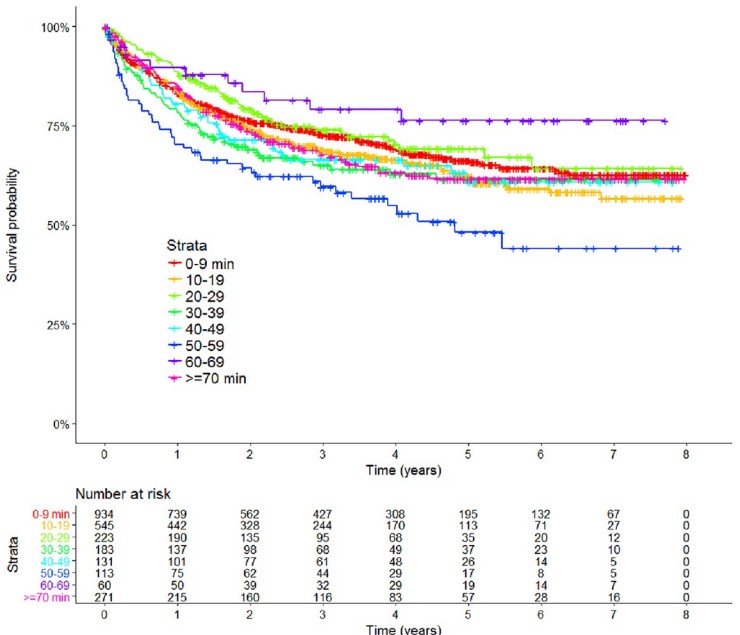

## Rectal cancer

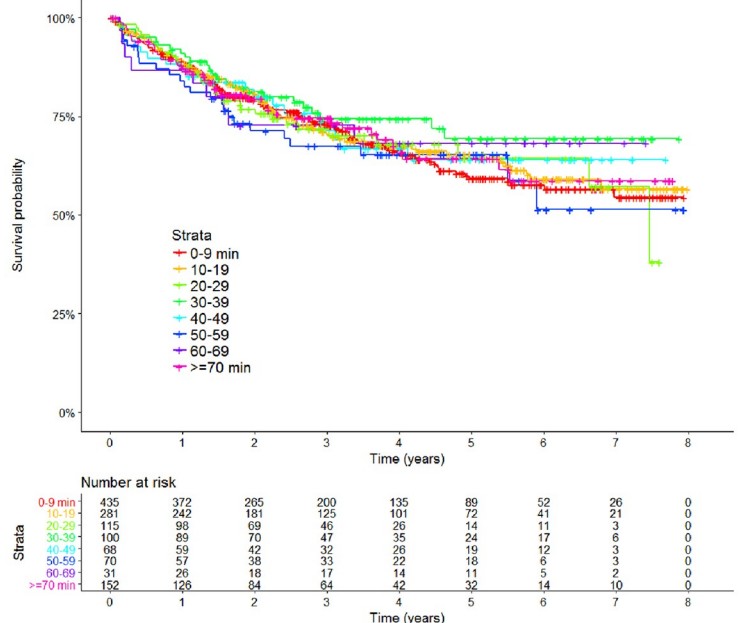

**Fig 3. Kaplan-Meier plots for cause-specific survival for patients with colon and rectal cancer respectively by travel time to the nearest hospital.**

**Table 3. Hazard ratios of cause-specific survival for operated patients estimated in a multiple cox regression analysis; stratified by sex and age at diagnosis (10-year groups) and adjusted for educational level, cohabiting status, elective/emergency surgery and tumour stage.**

|  | Colon Cancer | | Rectal Cancer | |
|---|---|---|---|---|
|  | HR | 95% CI | HR | 95% CI |
| **Travel time** | 0.999 | 0.997–1.002 | 0.997 | 0.992–1.002 |
| **Education level** |  |  |  |  |
| Low (ref) | 1 (ref) |  | 1 |  |
| Medium | 0.96 | 0.78–1.17 | 0.86 | 0.60–1.002 |
| Higher | 0.87 | 0.67–1.14 | 1.05 | 0.66–1.66 |
| **Cohabitation status** |  |  |  |  |
| Living alone (ref) | 1(ref) |  | 1 |  |
| Not living alone | 0.77 | 0.64–0.93 | 0.77 | 0.56–1.07 |
| **Operation** |  |  |  |  |
| Elective (ref) | 1 (ref) |  | 1 |  |
| Emergency | 2.66 | 2.21–3.20 | 5.5 | 2.69–11.3 |
| **Tumour stage** |  |  |  |  |
| I (ref) | 1 (ref) |  | 1 |  |
| II | 1.66 | 0.94–2.91 | 2.76 | 1.44–5.26 |
| III | 6.15 | 3.63–10.4 | 3.98 | 2.13–7.42 |
| IV | 22.9 | 13.5–38.9 | 20.7 | 10.8–39.7 |

to most other studies, we found no association between travel time and tumour stage at diagnosis [4, 5, 8, 26]. This finding might explain our study's overall absence of associations between travel time and survival.

Corroborating results from earlier studies, we observed differences between rural and urban patients with respect to age and socioeconomic factors. However, these differences did not appear to negatively influence the survival for the patients with longer distances to care.

There are factors in the organisation of the health care system in the Northern Health Care Region in Sweden, which may mitigate health care disparities associated with longer distance to care:

The national health care system in Sweden is tax funded and provides care to all residents at low out-of-pocket cost [27]. In addition, all patients in Sweden are entitled to free or subsidized travel to care. All 13 hospitals in the Northern Health Care Region are publicly owned and all have facilities to diagnose colorectal cancer. Each local hospital offers service to the population in its catchment area. The population, which is served per hospital, varies from approximately 25000 to 160000 [14]. Surgical treatment, especially for rectal cancer, has however been centralised to fewer hospitals in the Region during the study period. By the end of the study period, rectal cancer surgery was performed at only five hospitals.

Chemotherapy is given at most local hospitals under guidance from the only Oncology departments in the Region, located in Umeå and Sundsvall. In summary, all colorectal cancer patients in the Region can go to their local hospital, at a low cost, for diagnosis and in most cases also for treatment.

In a non-universal health care setting, socioeconomic differences between urban and rural populations could affect the access to care more than the distance to care itself. However, many of the previous studies in this research field from the U.S. adjusted for socioeconomy as a confounding factor, but still reported associations between longer travel time and worse colorectal cancer outcomes [2, 4, 8].

Associations between longer distance to care and worse survival have also been reported in European countries with national universal health care systems such as France or the U.K. [5, 23]. Thus, true distance-related barriers, not confounded by differences in the patient´s socioeconomy, are probably also present and important in countries with universal health care.

Subsidized travel to care could be one way to mitigate distance related barriers to care. A study from Norway, with the same population pattern and health care system as Sweden, found no association between travel distance and cancer survival [28]. In both Sweden and Norway, all patients are entitled to free or subsidized travel to care [29, 30]. In other universal (e.g., U.K.) or mixed health care systems (e.g., Australia), support with travel costs are based on income and/or distance to the caregiver [31, 32]. The potential role of free or subsidized travel deserves more attention, especially with regard to patient adherence to repeated oncological treatment and outcome.

From a health care system perspective, another distance-related barrier to care could be difficulties for rural GPs to refer their patients to hospitals in urban areas. The facilitation of swift referrals from GPs is one of the concepts in so-called 'rapid cancer diagnostic and assessment pathways' in the U.K. or 'standardised cancer care pathways in Sweden [33, 34]. During our study period, neither standardised cancer care pathways, nor colorectal screening had been implemented in the studied region. Travel time studies from the U.K. (Scotland) with implemented cancer pathways still show an association between travel time and poorer cancer outcomes [3].

Centralising care with the intention of improving the standard of care can also introduce a barrier to care—*if* travel time is associated with poorer outcomes [25]. As mentioned above, there has been a trend towards centralising surgery to fewer hospitals in the Northern Health Care Region, especially for rectal cancer. However, in one of the settings in our sensitivity analysis, we analysed travel time to the operating hospital rather than the nearest hospital. The results were the same: no association between travel time and survival was found.

Finally, there might be other factors—not related to the health care system—which explains the good outcome in Northern Sweden. As suggested in a Norwegian study, individuals living in rural areas might be more used to travel far in their everyday life—thus reducing travel distance as a barrier to care [28].

There are some methodological differences between our study and studies reporting an association between travel distance and survival. First, we measure travel time to care, whereas travel distance has been more commonly studied [5, 8, 23, 24]. Most studies, however, define place of care similarly to our study–i.e., the nearest hospital–but a few studies have also measured distance to a general practitioner (GP) [3, 26]. Furthermore, we used Geographical Information System (GIS) technology to measure individual travel time by car. In contrast, some studies use straight line or great circle approximations of distances from place of residence to hospitals.

In our main survival analysis, we handled travel time as a continuous variable with 10-minute intervals rather than categorizing travel distance into, for example, quintiles or different cut-off values [3, 8, 24]. This was done to avoid any presumptions on what is a meaningful travel time difference. When we handled travel time as categorical variable ($< 1$ h vs. $> 1$ h) in our additional setting for the survival analysis, no association between travel time and survival was found.

All these methodological differences may limit comparability, although probably not the validity, of our results.

## Limitations

The relatively few patients with long travel time to the nearest hospital constitute a potential lack of statistical power and an important limitation in our study. Only 516/3718 ≈ 14% of the patients had > 1 h travel time to their nearest hospital. This reflects that most of the population in Northern Sweden are concentrated in and near the cities. However, the proportion of patients with > 1 h travel time is in line with or higher than most other studies [3, 5, 8, 23]. Another potential limitation is the validity on causes of death data. To minimize the impact of co-morbidity, we analysed cause-specific instead of all-cause mortality. The validity of the cause of death for malignant disease in the Swedish Causes of Death Registry is approximately 90%, but there might be differential bias between rural and urban areas [19] There may also be information bias due to missing data on stage of disease at time of diagnosis, as no information on stage was available for 300 of the studied patients (≈8%).

## Strengths

The main strength of this study is the use of the Risk North population-based approach, a strategy that enables a health care system perspective based on individual data linked to health care and sociodemographic registries. This linkage enables precise computerized measurement of travel time as well as individual-specific information on possible confounders for a proposed travel time–survival association. Thanks to Sweden's assignment of personal identity numbers, only three patients moving abroad were lost for follow-up in the survival analysis.

## Conclusions

In contrast to earlier results from studies conducted in variety of settings, we found no association between travel time to the nearest hospital and stage of disease at time of diagnosis and survival in colorectal cancer. Thus, in the geographical setting of the Northern Health Care Region in Sweden, travel time to care does not appear to negatively influence colorectal cancer outcomes. The results suggest that the Swedish health care system manages to equalize disparities associated with travel distance to care. Future studies should compare different health care systems to identify factors that facilitate access to care in rural areas.

## Supporting information

**S1 Fig. The Risk North database—Individual data from three national cancer quality registers, with linked information from other nationwide health care and demographic registers.**
(DOCX)

**S2 Fig. Mean age, educational level and proportion of colorectal patients living alone by travel time to the nearest hospital.**
(DOCX)

**S3 Fig. The patient´s education level vs. travel time to their nearest hospital.**
(DOCX)

**S4 Fig. Proportion of patients living alone vs. travel time to their nearest hospital.**
(DOCX)

**S5 Fig. The patient´s tumour stage at diagnosis vs. travel time to the nearest hospital.**
(DOCX)

**S6 Fig. Proportion of elective or emergency surgery vs. travel time to the nearest hospital.**
(DOCX)

**S1 Table. Sensitivity analysis, results for analysing colorectal cancer as one entity.** Hazard ratios of cause-specific survival in colorectal cancer for operated patients estimated in a multiple cox regression analysis; stratified by sex and age at diagnosis (10-year groups) and adjusted for educational level, cohabiting status, elective/emergency surgery and tumour stage.
(DOCX)

**S2 Table. Sensitivity analysis, results for analysing all-cause survival.** Hazard ratios of all-cause survival for operated patients estimated in a multiple cox regression analysis; stratified by sex and age at diagnosis (10-year groups) and adjusted for educational level, cohabiting status, elective/emergency surgery and tumour stage.
(DOCX)

**S3 Table. Sensitivity analysis, results for analysing cause-specific survival without adjusting for tumour stage and emergency operations.** Hazard ratios of cause specific survival for operated patients estimated in a multiple cox regression analysis; stratified by sex and age at diagnosis (10-year groups) and adjusted for educational level and cohabiting status.
(DOCX)

**S4 Table. Sensitivity analysis, results for analysing travel time to the operating hospital.** Hazard ratios of cause-specific survival for operated patients estimated in a multiple cox regression analysis; stratified by sex and age at diagnosis (10-year groups) and adjusted for educational level, cohabiting status, elective/emergency surgery and tumour stage.
(DOCX)

**S5 Table. Sensitivity analysis, results when excluding patients with tumour stage IV.** Hazard ratios of cause-specific survival for operated patients estimated in a multiple cox regression analysis; stratified by sex and age at diagnosis (10-year groups) and adjusted for educational level, cohabiting status, elective/emergency surgery and tumour stage.
(DOCX)

**S6 Table. Additional setting of the multivariable regression survival analysis, travel time handled as a categorical variable; survival for patients with travel time < 1 h compared to patients traveling ≥1 h.** Hazard ratios of cause-specific survival for operated patients estimated in a multiple cox regression analysis; stratified by sex and age at diagnosis (10-year groups) and adjusted for educational level, cohabiting status, elective/emergency surgery and tumour stage.
(DOCX)

**S1 File. Regression analyses for travel time and survival.**
(DOCX)

## Acknowledgments

The authors would like to thank all included registries for their cooperation in enabling the creation of Risk North and this study. A special thanks to Mats Lambe for sharing his knowledge on register based research and giving feedback on this study.

## Author Contributions

**Conceptualization:** Olle Sjöström, Gustav Silander, Beatrice Melin.

**Data curation:** Anna M. Dahlin, Ingvar Syk, Beatrice Melin, Barbro Numan Hellquist.

**Formal analysis:** Olle Sjöström, Anna M. Dahlin, Barbro Numan Hellquist.

**Funding acquisition:** Olle Sjöström, Beatrice Melin.

**Investigation:** Olle Sjöström, Anna M. Dahlin, Beatrice Melin, Barbro Numan Hellquist.

**Methodology:** Olle Sjöström, Anna M. Dahlin, Beatrice Melin, Barbro Numan Hellquist.

**Project administration:** Olle Sjöström, Beatrice Melin, Barbro Numan Hellquist.

**Resources:** Beatrice Melin, Barbro Numan Hellquist.

**Software:** Anna M. Dahlin, Barbro Numan Hellquist.

**Supervision:** Beatrice Melin.

**Validation:** Olle Sjöström, Anna M. Dahlin, Ingvar Syk, Beatrice Melin.

**Visualization:** Olle Sjöström.

**Writing – original draft:** Olle Sjöström.

**Writing – review & editing:** Olle Sjöström, Anna M. Dahlin, Gustav Silander, Ingvar Syk, Beatrice Melin, Barbro Numan Hellquist.

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
