## [Decision Letter · Decision Letter 0]

17 Feb 2020

PONE-D-19-34718

Travel time to care does not affect survival for patients with colorectal cancer in northern Sweden

A data linkage study from the Risk North database

PLOS ONE

Dear Dr Sjöström,

Thank you for submitting your manuscript to PLOS ONE. After careful consideration, we feel that it has merit but does not fully meet PLOS ONE’s publication criteria as it currently stands. Therefore, we invite you to submit a revised version of the manuscript that addresses the points raised during the review process.

We would appreciate receiving your revised manuscript by Apr 02 2020 11:59PM. To enhance the reproducibility of your results, we recommend that if applicable you deposit your laboratory protocols in protocols.io, where a protocol can be assigned its own identifier (DOI) such that it can be cited independently in the future. For instructions see: http://journals.plos.org/plosone/s/submission-guidelines#loc-laboratory-protocols

We look forward to receiving your revised manuscript.

Kind regards,

Jung Eun Lee

Academic Editor

PLOS ONE

Additional Editor Comments (if provided):

Figures should stand alone. Please provide values on figures and details in the figure legends. For example, figure 2 does not have information on %.

Please provide location, name and etc. in figure 1.

Only multivariate analysis results, which are main findings, can be presented in the abstract.

Please provide characteristics (age, tumor stage, SES, surgery, cohabiting status, tumor location and etc.) of participants according to travel time (e.g. whether they were older with longer travel time?) instead of figures (cannot see well in the figures).

Did authors find a dose-response relationship between travel time and survival? Please state whether the association was linear. Please also provide results from quantile (quartile?) or categorical analysis.

On page 23, If authors believe that other studies were affected by confounding factors such as age, SES, and co-morbidity, please describe the limitation of each study in detail. (e.g. those studies did not adjust for or measure these factors? If they had adjusted for SES, would they have observed the null?)

On page 24, In the UK and France, positive or inverse associations were reported? Whenever authors reported the association in the previous studies, please state the direction of the association.

Journal Requirements:

2. In the ethics statement in the manuscript and in the online submission form, please provide additional information about the patient records used in your retrospective study, including: a) whether all data were fully anonymized before you accessed them and whether the data in the Risk North Database is publicly available and b) the date range (month and year) during which patients' medical records were accessed. If patients provided informed written consent to have data from their medical records used in research, please include this information.

3. To comply with PLOS ONE submission guidelines, in your Methods section, please provide additional information regarding your statistical analyses. For more information on PLOS ONE's expectations for statistical reporting, please see https://journals.plos.org/plosone/s/submission-guidelines.#loc-statistical-reporting.

5. We note that Figure 1 in your submission contains map images which may be copyrighted. All PLOS content is published under the Creative Commons Attribution License (CC BY 4.0), which means that the manuscript, images, and Supporting Information files will be freely available online, and any third party is permitted to access, download, copy, distribute, and use these materials in any way, even commercially, with proper attribution. For these reasons, we cannot publish previously copyrighted maps or satellite images created using proprietary data, such as Google software (Google Maps, Street View, and Earth). For more information, see our copyright guidelines: http://journals.plos.org/plosone/s/licenses-and-copyright.

You may seek permission from the original copyright holder of Figure 1 to publish the content specifically under the CC BY 4.0 license. 

If you are unable to obtain permission from the original copyright holder to publish these figures under the CC BY 4.0 license or if the copyright holder’s requirements are incompatible with the CC BY 4.0 license, please either i) remove the figure or ii) supply a replacement figure that complies with the CC BY 4.0 license. Please check copyright information on all replacement figures and update the figure caption with source information. If applicable, please specify in the figure caption text when a figure is similar but not identical to the original image and is therefore for illustrative purposes only.

Reviewers' comments:

Reviewer's Responses to Questions

**Comments to the Author**

1. Is the manuscript technically sound, and do the data support the conclusions?

Reviewer #1: Partly

Reviewer #2: Partly

2. Has the statistical analysis been performed appropriately and rigorously? 

Reviewer #1: No

Reviewer #2: I Don't Know

3. Have the authors made all data underlying the findings in their manuscript fully available?

Reviewer #1: Yes

Reviewer #2: Yes

4. Is the manuscript presented in an intelligible fashion and written in standard English?

Reviewer #1: Yes

Reviewer #2: Yes

5. Review Comments to the Author

Reviewer #1: This is an important contribution to the understanding of how access to services affects cancer outcomes. I have some queries and some concerns. The main concern is that the usual way of presenting data is that travel time is the independent variable and the outcome variable (survival, emergency surgery, stage etc) is the dependent variable. This is how the different curves in the Kaplan Meier plot are defined in Figure 3. However, in the tables the mean travel time is given as the dependent variable. I would expect to see, for example, what proportion of patients travelling >50 minutes required emergency surgery. This would be consistent with the other studies of this topic that were cited.

I am not surprised that the K-M curves for rectal cancer are separated less then for colonic cancer. However the survival curve for 50-59 minutes appears to be an outlier. It would be good to see the 95% confidence interval of this curve in particular.

There appear to be two different issues in the question of access. One is the distance decay effect seen in the studies cited by the authors, the other is the way in which distance is compensated in citizens' approach to symptoms. This factor is likely to be relevant in Northern Sweden where the overall population density is very low but there is concentration around a few settlements. In this context, it is important to note that Kravdal (ref 25) found: "most notably, those who lived in Oslo and Southern Norway had a relatively poor survival, given the size of the nearest hospital." If Southern Norway has the same characteristics as England & Scotland (and Denmark) and the rest of Norway resembles Northern Sweden, this would be an instructive finding in improving cancer services. Can the authors address this in heir discussion?

Reviewer #2: This is an interesting manuscript about the travel time and outcomes for cancer patients in Northern Sweden. There are several aspects of this manuscript that dampen my enthusiasm for this manuscript, as described below.

1. There needs to be a lot more detail on the setting of this study, as I suspect this is quite different than many other care delivery settings. Specifically, I would like to understand how the population density compares to other countries. I have a hard time knowing how 900,000 inhabitants in this area related to other areas. This seems very sparsely populated. Do many of these people live in small cities, is the distribution for across the area? It sounds as if the majority of the patients in the sample may be in what many other regions would consider rural or low volume settings. We also need information about availability of healthcare facilities and structure of these locations. I would want to know about both the cancer care delivery locations and access to ER/critical access centers. It would also be helpful to talk more about access in Sweden. Do most people go to a major surgery center and do they also get chemotherapy there? Without more information on context, this manuscript could be taken out of the appropriate context and difficult to relate to other care settings.

2. I found the map to be somewhat confusing. Is the blue representing where the population of the study came from or just what would be that time to drive to the center. Are the crosses the health centers or cities? This could be a very useful map if there was a better sense of service locations of different types, population density and travel lines.

3. I do not understand the comment that “Only patients who were treated with surgery were included in the multivariable analysis, as we wanted to study the impact of emergency compared to elective surgery.” Most of these patients would likely have an elective colon operation? I was very confused by this selection criteria and how it was applied, and why. If you are only looking at people who are diagnosed due to an emergency (obstruction) this would be a very different population that the common colon and rectal cancer patient. In addition, you have many stages so I am not sure if I am just not understanding this.

4. It would be helpful to write in the manuscript the percent in each category, not only the category of less than 10 minutes. What you really want to understand are those who travel greater distances. This does not give the reader a sense of how you conceptualize the “longer travel” group. The authors should justify the use of these intervals and why this makes clinical sense. I am not sure that these categories reflect how patients and providers think about distance. What is a meaningful time difference?

5. In the results, please explain better the association. A p-value tells me little. I want to know for how much difference in travel time did you see what difference in age. Please add numbers to give context.

6. This needs a sensitivity analysis or consideration of a categorical variable for travel time as the primary focus because the travel time is not linear. Conceptually, this also doesn’t make sense. You want to understand the difference between those who travel far vs. not, rather than the difference between 10 minutes of travel. This difference means something entirely different if you are talking 10 vs. 20 minutes or 90 vs. 100 minutes.

7. These units are challenging. I wonder if this is part of why the findings are as such because of the consideration of this variable as linear. When I look at figures, the authors do categorize better, but this is not coming through in the manuscript. I also would like to know how many people fall in these categories-the histogram is not labeled in a way that would correlate with the KM curves. I would also consider smaller number of categories as the numbers get very small in the survival analysis, which can lead to difficulty interpreting results.

8. There have been other recent studies that look at time traveled to care and time to other hospitals, very similar to this study. Please soften this language or refer to the other studies that use time.

9. Please discuss why you think the universal health care system might tie into your results with some literature backing. This is a very interesting part of the discussion.

6. PLOS authors have the option to publish the peer review history of their article (what does this mean?). If published, this will include your full peer review and any attached files.

Reviewer #1: No

Reviewer #2: No

---

## [Author Response · Author response to Decision Letter 0]

5 May 2020

Dear Editor!

Thank you for your comments and suggestions to improve our research and manuscript, we have successfully been able to adjust the manuscript on several points and now feel that it has improved. We now hope that it is possible to accept in its current revised form. 

Our responds are given point by point below and highlighted in the manuscript. 

Best Wishes, Olle Sjöström, Corresponding author.

Additional Editor Comments (if provided):

Figures should stand alone. Please provide values on figures and details in the figure legends. For example, figure 2 does not have information on %.

Thank you for this comment. We have added figures and values accordingly. Please see the revised figures.

Please provide location, name and etc. in figure 1.

Please see the new Figure 1. The travel time isochrones has been removed and replaced with name and location of the hospitals.

Only multivariate analysis results, which are main findings, can be presented in the abstract.

Please see the revised abstract, we have removed all results other results:

Results

No association between travel time and survival was observed, either in univariable analysis (colon HR 1.00 [95% CI 0.998 - 1.003]; rectal HR 0.998; [95% CI 0.995 -1.002]) or in multivariable Cox regression analysis (colon HR 0.999 [95% CI 0.997 - 1.002]; rectal HR 0.997 [95% CI 0.992- 1.002]). 

Please provide characteristics (age, tumor stage, SES, surgery, cohabiting status, tumor location and etc.) of participants according to travel time (e.g. whether they were older with longer travel time?) instead of figures (cannot see well in the figures).

Please see the revised Table 2., where we have added the variables. 

Did authors find a dose-response relationship between travel time and survival? Please state whether the association was linear. Please also provide results from quantile (quartile?) or categorical analysis.

No dose-response relationship between travel time and survival was seen, please see K-M curves in Figure 3. We have now also included an additional categorical analysis comparing survival for patients with travel time < 1 vs. patients traveling > 1 h. Please see the revised statistical analysis section and the result of the new categorical analysis:

In the additional multivariable regression survival analysis where travel time was handled as a categorical variable (survival for patients with travel time < 1 h was compared to patients traveling > 1 h) we found no association between travel time and survival, either for patients with colon cancer

HR 0.92[95% CI 0.71 - 1.19] or rectal cancer HR 0.84 [95% CI 0.52- 1.36]

On page 23, If authors believe that other studies were affected by confounding factors such as age, SES, and co-morbidity, please describe the limitation of each study in detail. (e.g. those studies did not adjust for or measure these factors? If they had adjusted for SES, would they have observed the null?)

Please see the revised section of the discussion.

However, many of the previous studies in this research field from the U.S. adjusted for socioeconomy as a confounding factor, but still reported associations between longer travel time and worse colorectal cancer outcomes 2,4,8 

On page 24, In the UK and France, positive or inverse associations were reported? Whenever authors reported the association in the previous studies, please state the direction of the association.

Please see the revised section of the discussion.

Associations between longer distance to care and worse survival have also been reported in European countries with national universal health care systems such as France or the U.K. 5,21

Journal Requirements:

The requirements have been met in the resubmission.

2. In the ethics statement in the manuscript and in the online submission form, please provide additional information about the patient records used in your retrospective study, including: a) whether all data were fully anonymized before you accessed them and whether the data in the Risk North Database is publicly available and b) the date range (month and year) during which patients' medical records were accessed. If patients provided informed written consent to have data from their medical records used in research, please include this information.

Please see the revised version methods and ethics sections.

Ethics

The Regional Board of Ethics in Umeå approved the design of the Risk North database and the present research project. The patients did not provide informed written consent but all data were fully anonymised before access. All colorectal cancer patients in Sweden are informed about registration in the SCRCR (i.e. the primary source of data in the present study) and an opt-out procedure for registration is used in the SCRCR. The data in the Risk North Database is not publicly available according to the Swedish data protection law. 

3. To comply with PLOS ONE submission guidelines, in your Methods section, please provide additional information regarding your statistical analyses. For more information on PLOS ONE's expectations for statistical reporting, please see https://journals.plos.org/plosone/s/submission-guidelines.#loc-statistical-reporting.

Please see the revised statistical analysis section. We have added more information on proportion of distribution, p-values in the manuscript and the full regression analysis as a supplementary file

Please see the revised cover letter. 

Detailed data is not possible to share without a scientific collaboration. As the RISK North database includes many variables that makes a person potentially identifiable we cannot share data publically on a PLOS ONE home page according to the Swedish data protection law. If researchers would like to get access to repeat the analyses access request can be at rccnorr@umu.se

5. We note that Figure 1 in your submission contains map images which may be copyrighted. All PLOS content is published under the Creative Commons Attribution License (CC BY 4.0), which means that the manuscript, images, and Supporting Information files will be freely available online, and any third party is permitted to access, download, copy, distribute, and use these materials in any way, even commercially, with proper attribution. For these reasons, we cannot publish previously copyrighted maps or satellite images created using proprietary data, such as Google software (Google Maps, Street View, and Earth). For more information, see our copyright guidelines: http://journals.plos.org/plosone/s/licenses-and-copyright.

Please see the new Figure 1, which created under CCO license. (Source: The Swedish Mapping Authority (Lantmäteriet), available according to open data license Creative Commons, CC0) 

Reviewer #1: This is an important contribution to the understanding of how access to services affects cancer outcomes. I have some queries and some concerns. The main concern is that the usual way of presenting data is that travel time is the independent variable and the outcome variable (survival, emergency surgery, stage etc) is the dependent variable. This is how the different curves in the Kaplan Meier plot are defined in Figure 3. However, in the tables the mean travel time is given as the dependent variable. I would expect to see, for example, what proportion of patients travelling >50 minutes required emergency surgery. This would be consistent with the other studies of this topic that were cited.

Please see the revised Table 2. The table now presents travel time as the independent variable, defined in 10 minutes intervals.

I am not surprised that the K-M curves for rectal cancer are separated less then for colonic cancer. However the survival curve for 50-59 minutes appears to be an outlier. It would be good to see the 95% confidence interval of this curve in particular.

We believe that Figure 3 would be to cluttered if 95% CI curves were included. The main point from the K-M curves in Figure 3 is that there is no obvious association between travel time and survival. 

There appear to be two different issues in the question of access. One is the distance decay effect seen in the studies cited by the authors, the other is the way in which distance is compensated in citizens' approach to symptoms. This factor is likely to be relevant in Northern Sweden where the overall population density is very low but there is concentration around a few settlements. In this context, it is important to note that Kravdal (ref 25) found: "most notably, those who lived in Oslo and Southern Norway had a relatively poor survival, given the size of the nearest hospital." If Southern Norway has the same characteristics as England & Scotland (and Denmark) and the rest of Norway resembles Northern Sweden, this would be an instructive finding in improving cancer services. Can the authors address this in heir discussion?

Regrettably, we have had some difficulties to understand this comment, since our study does not include hospital size as a variable. We have added information in the introduction and in the discussion in order to better describe the setting of this study.

Section in the introduction:

The Northern Healthcare Region has approximately 900 000 inhabitants living in an area of 224 000 km2, resulting in a population density of only 4 pop./km2. (U.K. 274 pop./km2, France 123 pop./km2) . The Region includes many rural areas where patients have to travel long distances to reach the nearest hospital (Figure 1).

Section in the discussion:

There are factors in the organisation of the health care system in the Northern Health Care Region in Sweden, which may mitigate health care disparities associated with longer distance to care:

The national health care system in Sweden is tax funded and provides care to all residents at low out-of-pocket cost.25 . In addition, all patients in Sweden are entitled to free or subsidized travel to care. All hospitals in the Northern Health Care Region are publicly owned and all have facilities to diagnose colorectal cancer. Surgical treatment, especially for rectal cancer, has however been centralised to fewer hospitals in the Region during the study period. Chemotherapy is given at most local hospitals under guidance from the only Oncology departments in the Region, located in Umeå and Sundsvall. In summary, all colorectal cancer patients in the Region can go to their local hospital, at a low cost, for diagnosis and in most cases also for treatment.

Reviewer #2: This is an interesting manuscript about the travel time and outcomes for cancer patients in Northern Sweden. There are several aspects of this manuscript that dampen my enthusiasm for this manuscript, as described below.

1. There needs to be a lot more detail on the setting of this study, as I suspect this is quite different than many other care delivery settings. Specifically, I would like to understand how the population density compares to other countries. I have a hard time knowing how 900,000 inhabitants in this area related to other areas. This seems very sparsely populated. Do many of these people live in small cities, is the distribution for across the area? It sounds as if the majority of the patients in the sample may be in what many other regions would consider rural or low volume settings. We also need information about availability of healthcare facilities and structure of these locations. I would want to know about both the cancer care delivery locations and access to ER/critical access centers. It would also be helpful to talk more about access in Sweden. Do most people go to a major surgery center and do they also get chemotherapy there? Without more information on context, this manuscript could be taken out of the appropriate context and difficult to relate to other care settings.

We have added information in the introduction and in the discussion in order to better describe the setting of this study. See above.

2. I found the map to be somewhat confusing. Is the blue representing where the population of the study came from or just what would be that time to drive to the center. Are the crosses the health centers or cities? This could be a very useful map if there was a better sense of service locations of different types, population density and travel lines.

Please see the new Figure 1, the travel time isochrones are removed. Instead the figure just shows the location of the hospitals.

3. I do not understand the comment that “Only patients who were treated with surgery were included in the multivariable analysis, as we wanted to study the impact of emergency compared to elective surgery.” Most of these patients would likely have an elective colon operation? I was very confused by this selection criteria and how it was applied, and why. If you are only looking at people who are diagnosed due to an emergency (obstruction) this would be a very different population that the common colon and rectal cancer patient. In addition, you have many stages so I am not sure if I am just not understanding this.

Please see the revised data analysis section, for hopefully a better explanation.

Patients were excluded if data were missing for any of the co-variables in the multivariable analysis, and consequently we excluded all non-operated patients (missing data on the variable operation).

4. It would be helpful to write in the manuscript the percent in each category, not only the category of less than 10 minutes. What you really want to understand are those who travel greater distances. This does not give the reader a sense of how you conceptualize the “longer travel” group. The authors should justify the use of these intervals and why this makes clinical sense. I am not sure that these categories reflect how patients and providers think about distance. What is a meaningful time difference?

Please see the revised Table 2 and the revised section in the discussion regarding travel time as a continuous or a categorical variable. In the revised Table 2 the patient´s characteristic - mean age, level of education, co-habiting status, tumour stage at diagnosis and type of surgery – is stratified by travel time to the nearest hospital.

In our main survival analysis, we handled travel time as a continuous variable with 10-minute intervals rather than categorizing travel distance into, for example, quintiles or different cut-off values. This was done to avoid any presumptions on what is a meaningful travel time difference. 

Now we added an additional analysis where travel time is handled as a categorical variable (< 1 h vs. > 1 h) But still, no association between travel time and survival was found,.

5. In the results, please explain better the association. A p-value tells me little. I want to know for how much difference in travel time did you see what difference in age. Please add numbers to give context.

Please see the revised Table 2 – row mean age by different travel time.

6. This needs a sensitivity analysis or consideration of a categorical variable for travel time as the primary focus because the travel time is not linear. Conceptually, this also doesn’t make sense. You want to understand the difference between those who travel far vs. not, rather than the difference between 10 minutes of travel. This difference means something entirely different if you are talking 10 vs. 20 minutes or 90 vs. 100 minutes.

Please see the results for the additional analysis for travel time as a categorical variable, and answer to comment 4.

7. These units are challenging. I wonder if this is part of why the findings are as such because of the consideration of this variable as linear. When I look at figures, the authors do categorize better, but this is not coming through in the manuscript. I also would like to know how many people fall in these categories-the histogram is not labeled in a way that would correlate with the KM curves. I would also consider smaller number of categories as the numbers get very small in the survival analysis, which can lead to difficulty interpreting results.

Please see the revised Figure 2 with legend and Table 2.

8. There have been other recent studies that look at time traveled to care and time to other hospitals, very similar to this study. Please soften this language or refer to the other studies that use time.

Please see the revised section of the discussion.

First, we measure travel time to care, whereas travel distance has been more commonly studied

9. Please discuss why you think the universal health care system might tie into your results with some literature backing. This is a very interesting part of the discussion.

Please see the revised section of the discussion.

In a non-universal health care setting, socioeconomic differences between urban and rural populations could affect the access to care more than the distance to care itself. However, many of the previous studies in this research field from the U.S. adjusted for socioeconomy as a confounding factor, but still reported associations between longer travel time and worse colorectal cancer outcomes 2,4,8 

Associations between longer distance to care and worse survival have also been reported in European countries with national universal health care systems such as France or the U.K. 5,21 Thus, true distance-related barriers, not confounded by differences in the patient´s socioeconomy, are probably also present and important in countries with universal health care. 

Subsidized travel to care could be one way to mitigate distance related barriers to care. A study from Norway, with the same population pattern and health care system as Sweden, found no association between travel distance and cancer survival.26 In both Sweden and Norway, all patients are entitled to free or subsidized travel to care.27,28 In other universal (e.g., U.K.) or mixed health care systems (e.g., Australia), support with travel costs are based on income and/or distance to the caregiver. 29,30 The potential role of free or subsidized travel deserves more attention, especially with regard to patient adherence to repeated oncological treatment and outcome.

---

## [Decision Letter · Decision Letter 1]

22 Jun 2020

PONE-D-19-34718R1

Travel time to care does not affect survival for patients with colorectal cancer in northern Sweden

A data linkage study from the Risk North database

PLOS ONE

Dear Dr. Sjöström,

Thank you for submitting your manuscript to PLOS ONE. One reviewer pointed out that additional reason for no association could be discussed apart from the subsidizing system. Maybe can authors get some clues from Kravdal et al.?  

Please submit a revised version of the manuscript that addresses the points raised.

One reviewer's comments;

They do not offer a clear suggestion as to why this might be apart from indicating the subsidised travel for patients' attendance at hospital. My question is, do the Northern Swedish patients have good outcomes because they have a culture which enables them to compensate for their isolation? In this respect, is there anything to be learned from the study by Kradval (now ref 26) comparing the north-south geography of Norway with that of Sweden?

A few minor changes are required.

1) please provide a citation for  "The Northern Healthcare Region has approximately 900 000 inhabitants

living in an area of 224 000 km2 , resulting in a population density of only 4 pop./km2(U.K. 274 pop./km2 ,France 123 pop./km2)

2) please specify whether authors used a two-sided test when presenting a significance level of .05. 

3) In figure 2, please add tick marks on the X axis. For example, where is 110 min (n=110 (3%))? It is hard to figure it out exactly.  It is all the same color beyond 80-90 min.  Please also write "(minutes)" on the X axis label :Travel time (minutes)

4) In table 2, what statistical method was used to obtain p values for either continuous or categorical variable? 

Please submit your revised manuscript by July 10 2020 11:59PM. If you will need more time than this to complete your revisions, please reply to this message or contact the journal office at plosone@plos.org. Please include the following items when submitting your revised manuscript:

We look forward to receiving your revised manuscript.

Kind regards,

Jung Eun Lee

Academic Editor

PLOS ONE

Additional Editor Comments (if provided):

There are minor comments as follows.

Please specify statistical tests that authors used for new Table 2.

When authors discuss health care system in Sweden compared to other countries, please provide information on some statistics on health care delivery in Sweden to get clearer idea (e.g. population density, hospital facilities per population, fewer hospitals for surgical treatment for rectal cancer-> did authors have some numbers?).

Reviewers' comments:

Reviewer's Responses to Questions

**Comments to the Author**

1. If the authors have adequately addressed your comments raised in a previous round of review and you feel that this manuscript is now acceptable for publication, you may indicate that here to bypass the “Comments to the Author” section, enter your conflict of interest statement in the “Confidential to Editor” section, and submit your "Accept" recommendation.

Reviewer #1: (No Response)

Reviewer #2: All comments have been addressed

2. Is the manuscript technically sound, and do the data support the conclusions?

Reviewer #1: Yes

Reviewer #2: Yes

3. Has the statistical analysis been performed appropriately and rigorously? 

Reviewer #1: I Don't Know

Reviewer #2: Yes

4. Have the authors made all data underlying the findings in their manuscript fully available?

Reviewer #1: Yes

Reviewer #2: Yes

5. Is the manuscript presented in an intelligible fashion and written in standard English?

Reviewer #1: Yes

Reviewer #2: Yes

6. Review Comments to the Author

Reviewer #1: This is a much clearer presentation of the data and the points have all been addressed with the exception of one, where the authors did not understand my point! To express it differently, the authors show the lack of distance decay in outcomes from colorectal cancer and also point out that these findings differ from the consensus in other literature. They do not offer a clear suggestion as to why this might be apart from indicating the subsidised travel for patients' attendance at hospital. My question is, do the Northern Swedish patients have good outcomes because they have a culture which enables them to compensate for their isolation? In this respect, is there anything to be learned from the study by Kradval (now ref 26) comparing the north-south geography of Norway with that of Sweden?

Reviewer #2: The authors have addressed the concerns of the reviewers well. I appreciated the additional context about the population evaluated in this paper.

7. PLOS authors have the option to publish the peer review history of their article (what does this mean?). If published, this will include your full peer review and any attached files.

Reviewer #1: Yes: Dr S. Michael Crawford

Reviewer #2: No

---

## [Author Response · Author response to Decision Letter 1]

9 Jul 2020

Dear Editor!

Thank you for your further comments to the manuscript. We have adjusted the manuscript according to your suggestions. Hopefully the manuscript is now acceptable!

Our responds are given point-by-point below and highlighted in the manuscript. 

Best Wishes, Olle Sjöström, Corresponding author.

One reviewer's comments;

They do not offer a clear suggestion as to why this might be apart from indicating the subsidised travel for patients' attendance at hospital. My question is, do the Northern Swedish patients have good outcomes because they have a culture which enables them to compensate for their isolation? In this respect, is there anything to be learned from the study by Kradval (now ref 26) comparing the north-south geography of Norway with that of Sweden?

We agree with Dr S. Michael Crawford (Reviewer #1) that the reasons to the good outcome in Northern Sweden might be found in protective factors not related the health care system. Differences on a health care system level with possible effects on the association between survival and distance to care are however easier to discuss since references describing different health care systems are more available and distinct. Psychosocial factor would be more speculative to discuss as we do not have any data to support it.

In the study by Kravdal et al the authors speculate that individuals living in Northern Norway - due to psychological or social factors are more used to travel far in their everyday life - thus reducing travel distance as a barrier to care. They also discuss the possibility of a healthier lifestyle among the population outside the city centres - resulting in a more favourable prognosis. Neither our study nor the study by Kravdal et al have however any results to support these by all means much plausible hypothesises. 

But in order to broaden the discussion to factors outside the health care system we have added the following in the Discussion:

“ Finally, there might be other factors - not related to the health care system - which explains the good outcome in Northern Sweden. As suggested in a Norwegian study, individuals living in rural areas might be more used to travel far in their everyday life - thus reducing travel distance as a barrier to care.28 ”

A few minor changes are required.

1) please provide a citation for "The Northern Healthcare Region has approximately 900 000 inhabitants

living in an area of 224 000 km2 , resulting in a population density of only 4 pop./km2(U.K. 274 pop./km2 ,France 123 pop./km2)

Please see the revised part of the Introduction and new references 14 and 15. 

2) please specify whether authors used a two-sided test when presenting a significance level of .05. 

We have changed the paragraph “To test for proposed associations, we used two-tailed between-subject Student’s t-test for continuous parametric variables and Spearman’s test for non-parametric ranked variables (α=0.05)” to ”To test for proposed associations, we used two-sided between-subject Student’s t-test for continuous parametric variables and Spearman’s test for non-parametric ranked variables (α=0.05)”

3) In figure 2, please add tick marks on the X axis. For example, where is 110 min (n=110 (3%))? It is hard to figure it out exactly. It is all the same color beyond 80-90 min. Please also write "(minutes)" on the X axis label :Travel time (minutes)

We have revised Figure 2 according to your suggestions, which made the histogram much easier to understand!

4) In table 2, what statistical method was used to obtain p values for either continuous or categorical variable?

Please see the revised version of Table 2, footnotes indicating statistical method has been added.

Additional Editor Comments (if provided):

There are minor comments as follows.

Please specify statistical tests that authors used for new Table 2.

Please see 4) above! 

When authors discuss health care system in Sweden compared to other countries, please provide information on some statistics on health care delivery in Sweden to get clearer idea (e.g. population density, hospital facilities per population, fewer hospitals for surgical treatment for rectal cancer-> did authors have some numbers?).

Please see the revised section of the discussion:

”All 13 hospitals in the Northern Health Care Region are publicly owned and all have facilities to diagnose colorectal cancer. Each local hospital offers service to the population in its catchment area. The population, which is served per hospital, varies from approximately 25000 to 160000.14 Surgical treatment, especially for rectal cancer, has however been centralised to fewer hospitals in the Region during the study period. By the end of the study period, rectal cancer surgery was performed at only five hospitals. ”

Reviewers' comments:

Reviewer's Responses to Questions

Comments to the Author

1. If the authors have adequately addressed your comments raised in a previous round of review and you feel that this manuscript is now acceptable for publication, you may indicate that here to bypass the “Comments to the Author” section, enter your conflict of interest statement in the “Confidential to Editor” section, and submit your "Accept" recommendation.

Reviewer #1: (No Response)

Reviewer #2: All comments have been addressed

2. Is the manuscript technically sound, and do the data support the conclusions?

Reviewer #1: Yes

Reviewer #2: Yes

3. Has the statistical analysis been performed appropriately and rigorously? 

Reviewer #1: I Don't Know

Reviewer #2: Yes

4. Have the authors made all data underlying the findings in their manuscript fully available?

Reviewer #1: Yes

Reviewer #2: Yes

5. Is the manuscript presented in an intelligible fashion and written in standard English?

Reviewer #1: Yes

Reviewer #2: Yes

6. Review Comments to the Author

Reviewer #1: This is a much clearer presentation of the data and the points have all been addressed with the exception of one, where the authors did not understand my point! To express it differently, the authors show the lack of distance decay in outcomes from colorectal cancer and also point out that these findings differ from the consensus in other literature. They do not offer a clear suggestion as to why this might be apart from indicating the subsidised travel for patients' attendance at hospital. My question is, do the Northern Swedish patients have good outcomes because they have a culture which enables them to compensate for their isolation? In this respect, is there anything to be learned from the study by Kradval (now ref 26) comparing the north-south geography of Norway with that of Sweden?

Please see above!

Reviewer #2: The authors have addressed the concerns of the reviewers well. I appreciated the additional context about the population evaluated in this paper.

7. PLOS authors have the option to publish the peer review history of their article (what does this mean?). If published, this will include your full peer review and any attached files.

Do you want your identity to be public for this peer review? For information about this choice, including consent withdrawal, please see our Privacy Policy.

Reviewer #1: Yes: Dr S. Michael Crawford

Reviewer #2: No

---

## [Editor Report · Decision Letter 2]

15 Jul 2020

Travel time to care does not affect survival for patients with colorectal cancer in northern Sweden

A data linkage study from the Risk North database

PONE-D-19-34718R2

Dear Dr. Sjöström,

We’re pleased to inform you that your manuscript has been judged scientifically suitable for publication and will be formally accepted for publication once it meets all outstanding technical requirements.

Kind regards,

Jung Eun Lee

Academic Editor

PLOS ONE

---

## [Editor Report · Acceptance letter]

20 Jul 2020

PONE-D-19-34718R2 

Travel time to care does not affect survival for patients with colorectal cancer in northern Sweden
A data linkage study from the Risk North database 

Dear Dr. Sjöström:

I'm pleased to inform you that your manuscript has been deemed suitable for publication in PLOS ONE. Congratulations! Your manuscript is now with our production department. 

Kind regards, 

on behalf of

Dr. Jung Eun Lee 

Academic Editor

PLOS ONE